# Assessment of the Immune Response of Clinically Infected Calves to *Cryptosporidium parvum* Infection

**Wael El-Deeb** [1,2,*] [iD], **Olimpia Iacob** [3], **Mahmoud Fayez** [4,5] [iD], **Ibrahim Elsohaby** [6,7] [iD], **Abdulrahman Alhaider** [1], **Hermine V. Mkrtchyan** [8], **Abdelazim Ibrahim** [9,10] and **Naser Alhumam** [11]

1.  Department of Clinical Sciences, College of Veterinary Medicine, King Faisal University, Al-Ahsa 31982, Saudi Arabia
2.  Department of Internal Medicine, Infectious Diseases and Fish Diseases, Faculty of Veterinary Medicine, Mansoura University, Mansoura 35516, Egypt
3.  Clinics Department, Faculty of Veterinary Medicine, University of Agricultural Sciences and Veterinary Medicine, 700490 Iasi, Romania
4.  Al Ahsa Veterinary Diagnostic Lab, Ministry of Environment, Water and Agriculture, Al-Ahsa 31982, Saudi Arabia
5.  Department of Bacteriology, Veterinary Serum and Vaccine Research Institute, Ministry of Agriculture, Cairo 12618, Egypt
6.  Department of Animal Medicine, Faculty of Veterinary Medicine, Zagazig University, Zagazig 44511, Egypt
7.  Department of Infectious Diseases and Public Health, Jockey Club of Veterinary Medicine and Life Sciences, City University of Hong Kong, Kowloon 999077, Hong Kong
8.  School of Biomedical Sciences, University of West London, St Mary's Rd, London W5 5RF, UK
9.  Department of Pathology, College of Veterinary Medicine, King Faisal University, Al-Ahsa 31982, Saudi Arabia
10. Department of Pathology, College of Veterinary Medicine, Suez Canal University, Ismailia 41522, Egypt
11. Department of Microbiology and Parasitology, College of Veterinary Medicine, King Faisal University, Al-Ahsa 31982, Saudi Arabia
*   Correspondence: weldeeb@kfu.edu.sa

**Abstract:** *Cryptosporidium parvum* (*C. parvum*) infection is one of the main causes of diarrhea in calves. The current study assessed the role of blood biomarkers (acute-phase proteins (APPs), procalcitonin, neopterin, cytokines, and oxidative stress in the pathogenesis, diagnosis, and prognosis of calves naturally infected with *C. parvum*. Fifty-seven calves, aged from 10 to 45 days, were detected positive for *C. parvum* and were allotted into the diseased group; twenty healthy calves were selected as a control group. Serum amyloid A, haptoglobin, cytokines, neopterin, procalcitonin, and stress biomarkers were tested in the diseased and healthy groups. The serum levels of APPs, cytokines, neopterin, procalcitonin, and malondialdehyde increased, whereas antioxidant levels were significantly decreased in diseased calves compared to the healthy group. Moreover, all examined biomarkers were significantly altered towards normal values in infected calves following different treatment protocols. All biomarkers examined were highly effective in discriminating between *C. parvum*-infected calves and healthy individuals. Furthermore, the area under the curve (AUC) showed that all tested parameters had a higher degree of prognostic accuracy in predicting the treatment response of calves naturally infected with *C. parvum*. Our data suggest the usefulness of the examined biomarkers in the immune pathogenesis of the *C. parvum* infection in calves, contributing to diagnosis and treatment efficacy.

**Keywords:** procalcitonin; neopterin; cytokines; *Cryptosporidium*; haptoglobin; malondialdehyde

## 1. Introduction

Diarrhea remains a major cause of calf mortality, resulting in significant economic losses to the cattle breeding industry. Independent of the etiology or the pathogenicity mechanism, diarrhea induces loss of electrolytes and water through the feces and lowers milk intake. Acidosis associated with electrolyte imbalance (especially the decrease in

sodium concentration and the increase or decrease in potassium concentration) is accompanied by negative energy balance caused by anorexia and malabsorption [1].

*Cryptosporidium* spp. is a leading cause of diarrhea in newborn calves. This highly infectious apicomplexan protozoan parasite colonizes the apical surface of enterocytes, increasing the peristalsis of the small intestine and decreasing intestinal absorption [2]. Dixon et al. [3] showed that *Cryptosporidium* spp. had the highest incidences in calves aged 1–4 weeks. Of most, *Cryptosporidium parvum* (*C. parvum*) was responsible for 85% of infections in calves before weaning, inducing moderate to severe diarrhea with aqueous yellowish or mucoid feces [2]. In addition, many subtypes of *Cryptosporidium* spp. pose a risk of zoonotic transmission in humans, especially children [4].

Acute-phase proteins (APPs) are essential immune responses that help configure the extent of infection and eliminate damaging factors. The infection triggers the hepatic production of APPs mediated by the secretion of proinflammatory cytokines (CYTs). Proinflammatory cytokines have a short half-life span, whereas APPs can be detected for more extended periods. Numerous disease conditions, such as respiratory tract infections, urinary tract infections, metritis, mastitis, and infection with *Trypanosoma evansi*, have been associated with increased levels of APPs, such as haptoglobin (HP), serum amyloid A (major APPs), and fibrinogen in ruminants [5–11]. In bacterial and parasitic diseases, procalcitonin (PCT) is measured as a diagnostic marker because it can rapidly increase in circulation after the synthesis of CYTs [12]. The neopterin (NPT) marker is part of the cell-mediated immunity system derived from monocyte/macrophages [13]. The state of oxidative stress (OS) occurs once the balance between antioxidant and oxidant agents is changed, resulting from the overproduction of reactive oxygen species (ROS).

There are limited data about the immune response of calves against the clinical *Cryptosporidium* infection in the literature. Therefore, this work (a) investigated the relationships between acute-phase response and OS biomarkers and cryptosporidiosis in calves; (b) studied how these biomarkers relate to disease pathogenesis and immune response; and (c) assessed the capacity of these markers to differentiate between diseased and healthy calves and monitor treatment response.

## 2. Materials and Methods

### 2.1. Calf Enrollment

A total of 1688 Holstein calves, aged from 10 to 45 days, were investigated between May and August 2018 on one farm in the Middle region, Saudi Arabia. A total of 202 out of 1688 calves exhibited a sudden onset of pasty yellow to watery diarrhea or had soiled perineum and/or tail. Fecal samples were collected individually by rectal touch in sterile containers for specific parasitological examinations. Of the 202 diarrheic calves, 57 were found to be infected with *C. parvum*. Calves (*n* = 145) with clinical signs of diarrhea due to other causes or mixed infections with *Cryptosporidium* spp. were excluded from this study to avoid misclassification. Whole blood samples were collected from *C. parvum*-infected calves and the healthy control group (control = 20). The control calves were aged from 10 to 45 days and were free from cryptosporidiosis (negative fecal examination for oocysts and negative for antigen ELISA testing) or any other clinical problems. Blood samples were centrifuged, then serum-separated and stored at −20 °C for further biochemical analysis.

*C. parvum*-infected calves were treated with either paromomycin (*n* = 21 cases; 100 mg/Kg body weight for 10 consecutive days), halofuginone lactate (*n* = 22 cases; 0.1 mg/kg body weight for 7 consecutive days), or azithromycin (*n* = 14; 1500 mg/Kg body weight for 7 consecutive days). The treatment protocols were documented in previous reports [14–16]. Blood samples and feces were obtained seven days post-treatment from only 55 calves.

### 2.2. Fecal Sample Examination

Fecal samples were diluted volume per volume into the dilution buffer of the commercial sandwich ELISA kit (Multiscreen Antigen ELISA Calf digestive, Bio k348, Bio-X Diagnostics S.A., Rochefort, Belgium) for detection of the antigens of rotavirus, coron-

avirus, *E. coli*, and *Cryptosporidium* according to the kit instructions. Fecal samples positive for *Cryptosporidium* were further investigated for *Salmonella* spp. as described by Van Duijkeren et al. [17]. *Cryptosporidium* was confirmed in *Salmonella*-negative samples by demonstrating and quantitating *Cryptosporidium* oocysts using modified Ziehl Neelsen staining and flotation test (the FLOTAC method) as described previously [18,19].

### 2.3. Molecular Confirmation

*Cryptosporidium*-positive fecal samples were diluted 1:4 in phosphate-buffered saline and then subjected to total DNA extraction according to Morgan et al. [20] using the QIAamp DNA mini-kit (Qiagen S.A., Courtaboeuf, France) according to the manufacturer's instructions. According to Hadfield et al. [21], real-time PCR was carried out to amplify the SSU rRNA gene of the genus *Cryptosporidium* and LIB13, which is locus-specific for *C. parvum*.

### 2.4. Acute-Phase Proteins (APPs)

Test kits (Tridelta Development Ltd., Kildare, Ireland) were used to measure haptoglobin (HP) and serum amyloid A (SAA) in serum samples.

### 2.5. Proinflammatory Cytokines (CYTs)

In order to estimate CYTs concentrations (IL1-$\beta$, IFN-$\gamma$, and TNF-$\alpha$) in serum, commercial ELISAs (CUSABIO Biotech, Wuhan, China) were used according to the manufacturer's recommendations.

### 2.6. Procalcitonin and Neopterin

Our study used a commercial ELISA kit for measuring PCT concentration in cattle serum (Bovine Procalcitonin (PCT) ELISA Kit, CUSABIO Biotech, Wuhan, China) and an ELISA kit for measuring NPT concentrations in cattle serum (Bovine NPT (Neopterin) ELISA Kit, Fine Test, Wuhan Fine Biotech, Wuhan, China).

### 2.7. Oxidative Stress Biomarkers

A colorimetric method was used to determine the serum malondialdehyde levels (MDA), reduced glutathione levels (GSH), and superoxide dismutase levels (SOD) (Biodiagnostic, El Omraniya, Egypt) as described previously by Ohkawa et al. [20], Beutler [21], and Nishikimi et al. [22], respectively.

### 2.8. Histopathology

Severely sick calves (*n* = 2) were euthanized and subjected to full necropsy. Intestinal specimens were fixed in 10% neutral buffered formalin, and histopathological examination was carried out as described previously by Suvarna et al. [23].

### 2.9. Data Analysis

Statistical analysis was conducted using Prism 8 version 8.4.0 (GraphPad, La Jolla, CA, USA) software and Stata Statistical Software v.16 (Stata Corp, College Station, TX, USA). The APPs, CYTs, OS markers, PCT, and NPT distributions in the serum of diarrheic calves naturally infected with *C. parvum* and healthy control calves were assessed for normality using the Shapiro–Wilk test. The non-parametric Wilcoxon Mann–Whitney and Wilcoxon signed rank tests were used to assess the differences between diseased and healthy calves and between infected calves before and after treatment, respectively. A Spearman's rank correlation test was used to determine correlations among different parameters. A receiver operator characteristic (ROC) curve was applied to each parameter to assess its diagnostic and prognostic accuracy. The AUC (area under the curve) was calculated and the optimal cut-off was identified using the Youden index (= maximum [sensitivity + specificity-1]).

## 3. Results

A total of 57 calves were detected infected with *C. parvum*. Calves showed sudden onset of pasty yellow to watery diarrhea. Collected fecal samples showed a large number of *C. parvum* oocysts (Figure 1) using the Ziehl Neelsen stain method. Histopathological examination of the intestine (2 calves) showed that the lining epithelium of the colonic mucosa was lost and replaced with eosinophilic cellular and karyorechtic debris that was admixed with degenerated neutrophils. The blood vessels of the lamina propria were congested along with mild lymphocytic and neutrophilic infiltration. Numerous basophilic schizonts (2–4 μm in length) of *C. parvum* were attached to the epithelium's apical and luminal surfaces (Figure 2).

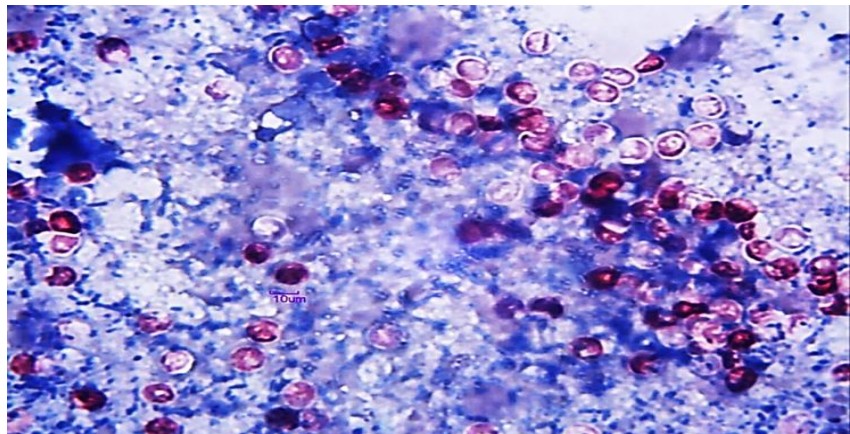

**Figure 1.** Fecal smear stained by the Ziehl Neelsen method. The smear shows a large number of oocysts belonging to the species *C. parvum*. The high density of oocysts confirms the cause of digestive disorders in the calves studied. The photography was carried out with the Leica DM 750 photon microscope at the ×1000 Immersion lens.

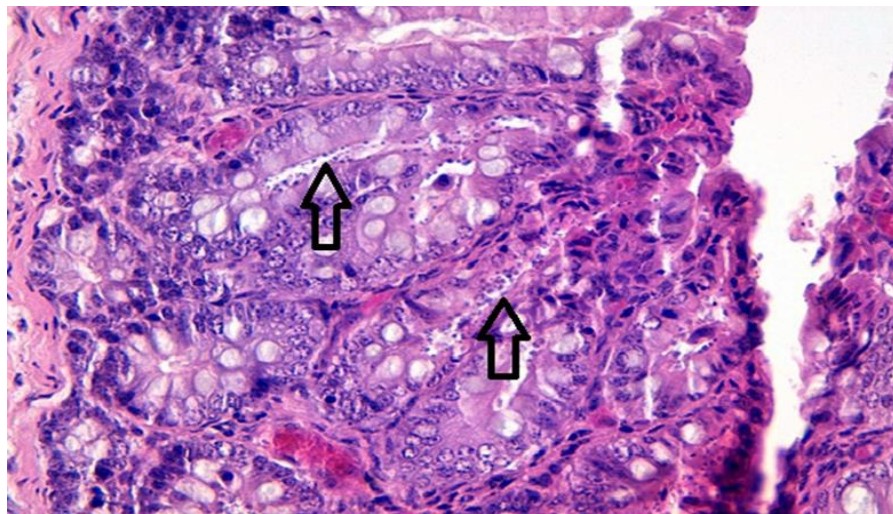

**Figure 2.** Cryptosporidial schizonts are attached to the intestinal gland epithelium's apical surface (arrows). H&E 400×.

The serum levels of APPs and CYTs in diarrheic calves with *C. parvum* were remarkably ($p < 0.001$) over those detected in healthy control calves (Figure 3), and all biomarkers showed a non-normal distribution. However, we found that diseased calves had a significantly lower level of antioxidant markers (GSH and SOD) compared to healthy calves (Figure 3). The serum levels of APPs, CYTs, PCT, and NPT as well as OS markers in calves with *C. parvum* pre- and post-treatment were measured in this study (Figure 4).

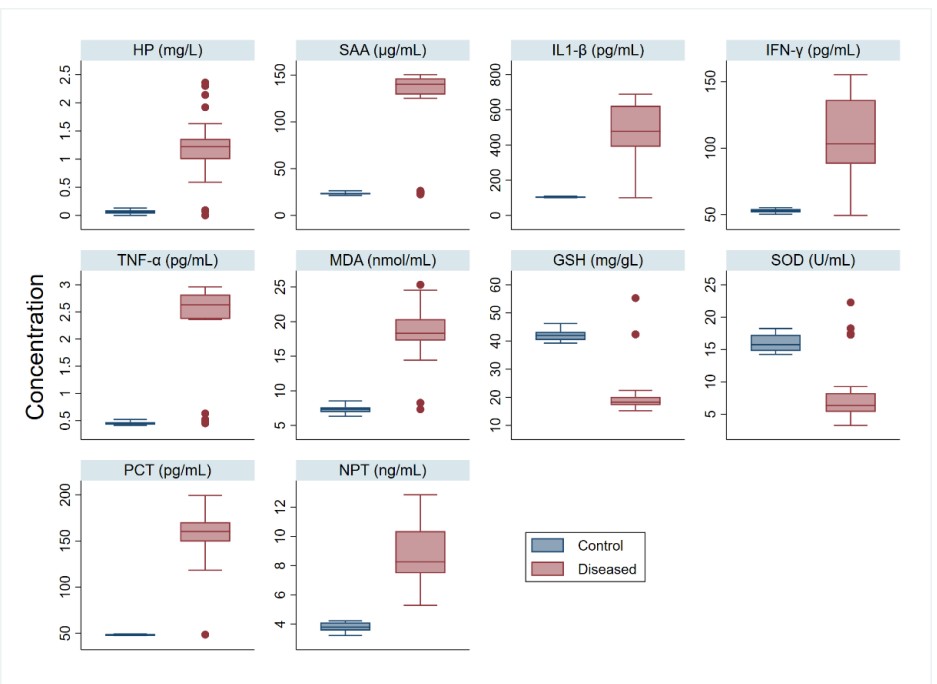

**Figure 3.** Box plots displaying the variability of acute-phase proteins (HP, Haptoglobin; SAA, serum amyloid A), proinflammatory cytokines (IL1-β, Interleukin one beta; IFN-γ, Interferon gamma; TNF-α, tumor necrosis factor alpha), oxidative stress markers (MDA, malondialdehyde; GSH, reduced glutathione; SOD, super oxide dismutase), procalcitonin (PCT), and neopterin (NPT) in the serum of diarrheic calves naturally infected with *C. parvum* (*n* = 57) and healthy control calves (*n* = 20). A difference with $p < 0.05$ was considered significant.

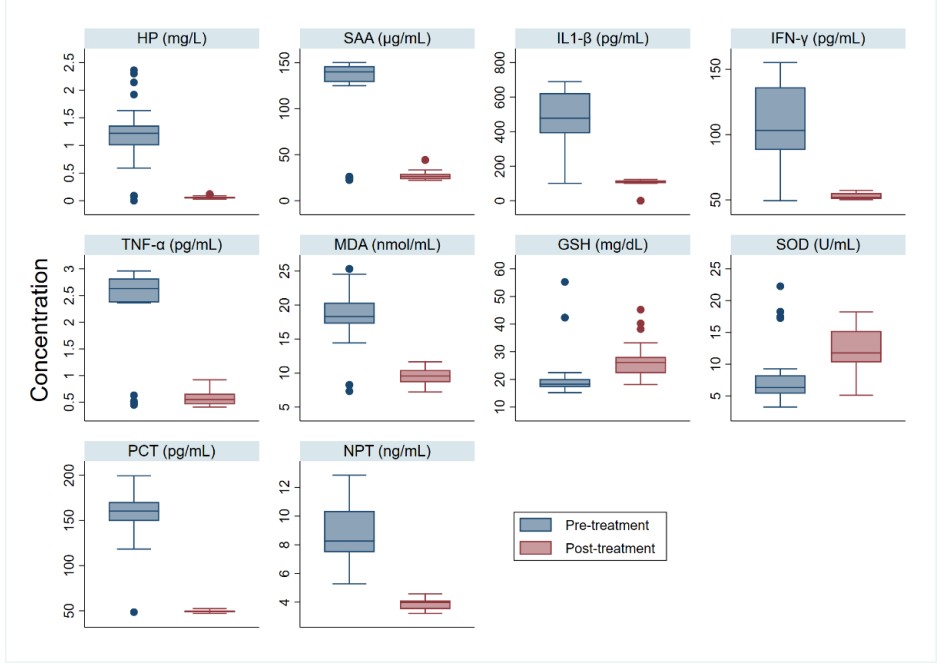

**Figure 4.** Box plots displaying variability of acute-phase proteins (HP, Haptoglobin; SAA, serum amyloid A), proinflammatory cytokines (IL1-β, Interleukin one beta; IFN-γ, Interferon gamma; TNF-α, tumor necrosis factor alpha), oxidative stress markers (MDA, malondialdehyde; GSH, reduced glutathione; SOD, super oxide dismutase), procalcitonin (PCT), and neopterin (NPT) in the serum of diarrheic calves naturally infected with *C. parvum* (*n* = 55) pre- and 7 days post-treatment. A difference with $p < 0.05$ was considered significant.

Moreover, a dramatic decline was detected in fecal oocyst shedding and serum levels of APPs and CYTs of infected calves after seven days of treatment, whereas the levels of antioxidant markers were significantly increased. Spearman's correlation results among the biomarkers in infected calves with *C. parvum* and healthy ones, as well as in infected calves before and after 7 days of treatment, are presented in Tables 1 and 2, respectively.

**Table 1.** Correlation matrix among acute-phase proteins (HP, Haptoglobin; SAA, serum amyloid A), proinflammatory cytokines (IL1-β, Interleukin one beta; IFN-γ, Interferon gamma; TNF-α, tumor necrosis factor alpha), oxidative stress markers (MDA, malondialdehyde; GSH, reduced glutathione; SOD, super oxide dismutase), procalcitonin (PCT), and neopterin (NPT) in the serum of diarrheic feedlot calves naturally infected with *C. parvum* and healthy calves.

| Parameters | Healthy/ Diseased | HP | SAA | IL-1β | IFN-γ | TNF-α | MDA | GSH | SOD | PCT | NPT |
|---|---|---|---|---|---|---|---|---|---|---|---|
| HP | 0.69 | 1.00 | | | | | | | | | |
| SAA | 0.68 | 0.45 | 1.00 | | | | | | | | |
| IL-1β | 0.62 | 0.64 | 0.54 | 1.00 | | | | | | | |
| IFN-γ | 0.66 | 0.61 | 0.64 | 0.72 | 1.00 | | | | | | |
| TNF-α | 0.70 | 0.66 | 0.35 | 0.52 | 0.54 | 1.00 | | | | | |
| MDA | 0.74 | 0.64 | 0.63 | 0.79 | 0.79 | 0.56 | 1.00 | | | | |
| GSH | −0.70 | −0.46 | −0.51 | −0.52 | −0.49 | −0.57 | −0.59 | 1.00 | | | |
| SOD | −0.57 | −0.63 | −0.40 | −0.51 | −0.45 | −0.49 | −0.48 | 0.33 | 1.00 | | |
| PCT | 0.75 | 0.39 | 0.49 | 0.38 | 0.42 | 0.50 | 0.46 | −0.55 | −0.23 | 1.00 | |
| NPT | 0.76 | 0.63 | 0.59 | 0.80 | 0.78 | 0.61 | 0.85 | −0.63 | −0.46 | 0.53 | 1.00 |

**Table 2.** Correlation matrix among acute-phase proteins (HP, Haptoglobin; SAA, serum amyloid A), proinflammatory cytokines (IL1-β, Interleukin one beta; IFN-γ, Interferon gamma; TNF-α, tumor necrosis factor alpha), oxidative stress markers (MDA, malondialdehyde; GSH, reduced glutathione; SOD, super oxide dismutase), procalcitonin (PCT), and neopterin (NPT) in the serum of diarrheic feedlot calves naturally infected with *C. parvum* pre- and 7 days post-treatment.

| Parameters | Pre/ Post-Treatment | HP | SAA | IL-1β | IFN-γ | TNF-α | MDA | GSH | SOD | PCT | NPT |
|---|---|---|---|---|---|---|---|---|---|---|---|
| HP | 0.80 | 1.00 | | | | | | | | | |
| SAA | 0.58 | 0.48 | 1.00 | | | | | | | | |
| IL-1β | 0.63 | 0.63 | 0.48 | 1.00 | | | | | | | |
| IFN-γ | 0.70 | 0.67 | 0.56 | 0.68 | 1.00 | | | | | | |
| TNF-α | 0.61 | 0.63 | 0.23 | 0.46 | 0.57 | 1.00 | | | | | |
| MDA | 0.76 | 0.69 | 0.53 | 0.70 | 0.76 | 0.56 | 1.00 | | | | |
| GSH | −0.64 | −0.53 | −0.42 | −0.51 | −0.48 | −0.51 | −0.65 | 1.00 | | | |
| SOD | −0.57 | 0.63 | −0.26 | −0.49 | −0.49 | −0.47 | −0.56 | 0.39 | 1.00 | | |
| PCT | 0.83 | 0.59 | 0.49 | 0.47 | 0.51 | 0.44 | 0.56 | −0.55 | −0.34 | 1.00 | |
| NPT | 0.85 | 0.73 | 0.54 | 0.73 | 0.76 | 0.58 | 0.85 | −0.63 | −0.53 | 0.70 | 1.00 |

The ROC curves were created (Figure 5), and AUC was estimated to assess the accuracy of each parameter in differentiating between diseased and healthy calves. Furthermore, the ROC curves and AUC results (Figure 6) showed that all tested parameters have a high degree of prognostic accuracy in predicting the treatment response of calves naturally infected with *C. parvum*.

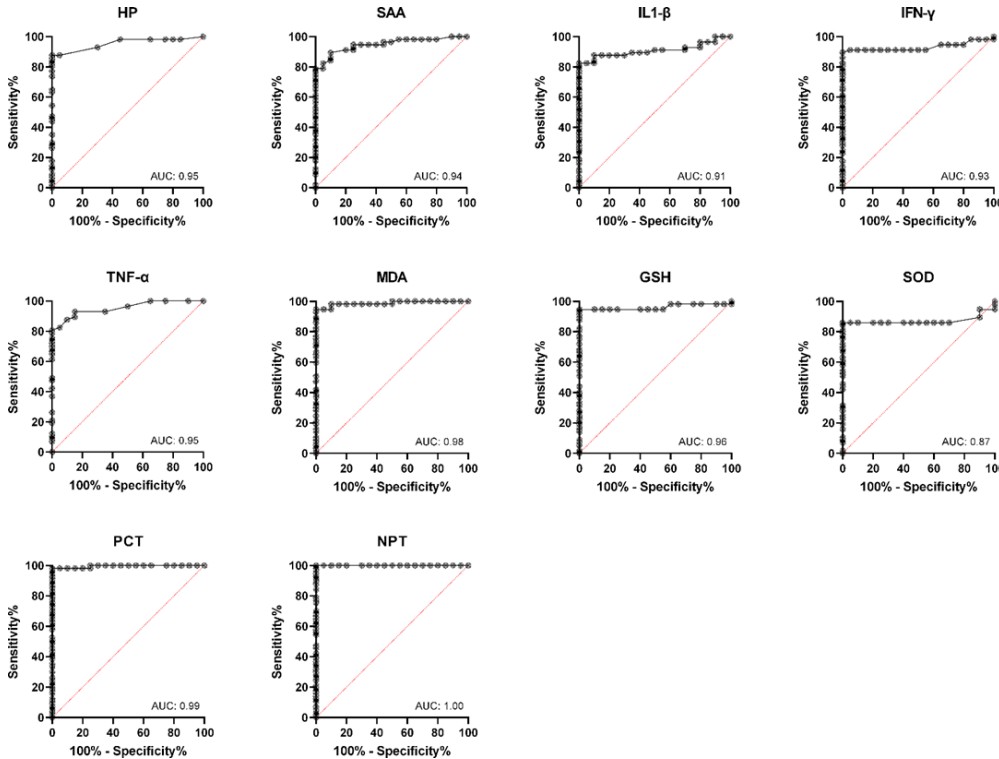

**Figure 5.** Receiver operating characteristic (ROC) curves for 10 selected blood biomarkers in diarrheic calves naturally infected with *C. parvum* and healthy control calves.

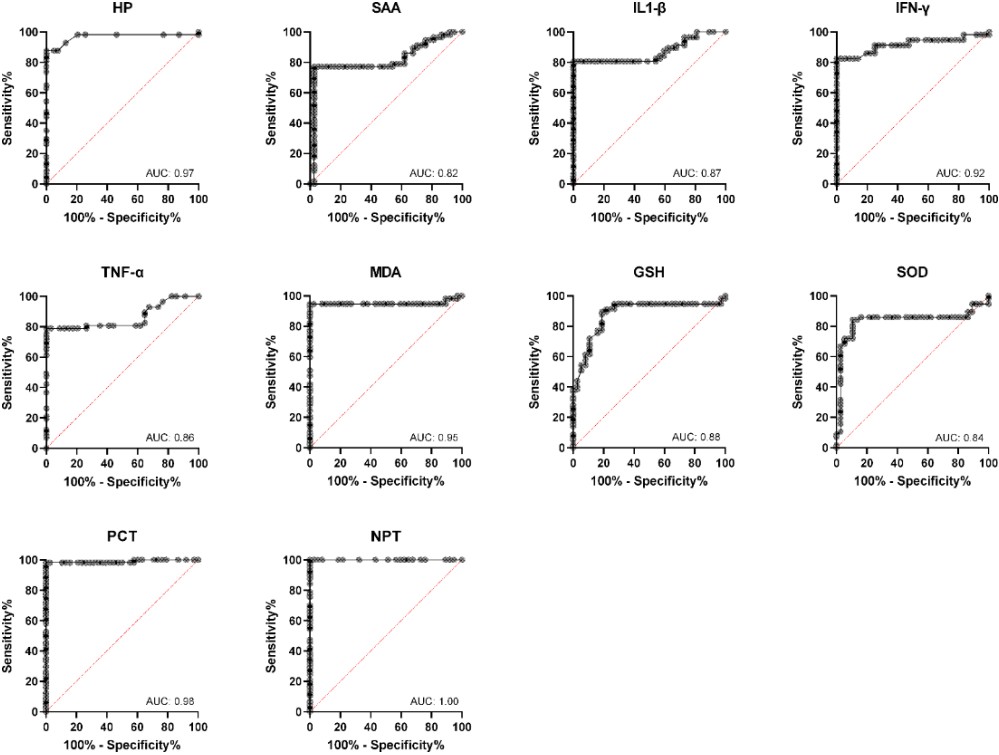

**Figure 6.** Receiver operating characteristic (ROC) curves for 10 selected blood biomarkers in diarrheic calves naturally infected with *C. parvum* pre- and 7 days post-treatment.

## 4. Discussion

*Cryptosporidium* is one of the leading causes of morbidity and mortality in calves [2,3]. The aim of this study was to examine blood changes in APPs, CYTs, and OS status in infected and recovered calves in order to highlight their role in *C. parvum* immunological pathogenesis and to evaluate their use as an additional tool for diagnosis and monitoring treatment response in infected calves. In this study, *C. parvum*-infected calves had higher serum HP and SAA levels, indicating a strong immune response to *Cryptosporidium* infections. High levels of HP can be mediated by the release of CYTs as a result of intestinal damage caused by infection. *Cryptosporidium* has previously been shown to cause severe damage to the intestinal lining epithelia by weakening microvilli and increasing mononuclear cell infiltration in the lamina propria [19]. The SAA response, on the other hand, can be supported because of its essential role in controlling animal immune defense during tissue injury. SAA's immunological effects are primarily mediated by stimulation of leukocyte migration and differentiation as well as increased IL-8 secretion [24–27]. Previous research has shown that the HP and SAA correlated with fecal scores are reliable indicators of the severity of diarrhea in Holstein dairy calves [28]. Furthermore, Hajimohammadi et al. [29] suggested that monitoring the APPs responses in calves with enteritis could be useful as a prognostic tool, thus aiding in treatment decisions. Similar findings have also been reported in lambs infected with *C. parvum* [30], goats infected with coccidiosis [31], and calves infected with *Eimeria zuernii* [32]. In contrast to our findings, Cenesiz et al. [33] found no differences in SAA levels between pretreatment and post-treatment in calves with *C. parvum* infection, which could be attributed to the severity of infection, age of examined calves, and time of blood sampling in infected calves. In addition, they examined a small number of cases (n = 10) where other causes of enteritis were not excluded.

CYTs levels in diseased calves were found to be higher than in control calves, indicating that calves with *C. parvum* infection have a significant acute-phase response as well as significant pathological changes. Previously, it was reported that the host immune response to *C. parvum* depends mainly on interferons [34,35]. Moreover, the activation of TLR4 receptors on dendritic cells by *C. parvum* infection leads to a number of cytokines being released, including IL-6, IL-1β, IL-12, IL-18, TNF-α, and interferons [36,37]. The current study revealed that CYTs play an important role in *Cryptosporidium* pathogenesis and immune response in infected animals. Our findings are consistent with previous reports of *Eimeria* spp. infection [38]. Previous studies reported that IL-1β and TNF-α are required for the stimulation of prostaglandin (PG) synthesis. In cases of intestinal cell lines infected with *Cryptosporidium* spp., higher blood levels of prostaglandins E2 and F2 as well as prostaglandin HS-2, a protein that controls them, were detected [39]. However, neither IL-1β nor TNF-α leads to enteritis; rather, it may occur as a result of PG altering the chloride secretion. PG also has a defensive role through increasing intestinal mucin expression, which can hinder sporozoite attachment to the intestinal mucosa [40,41]. IL-1 and TNF-mRNA have been found in the lamina propria of the intestine of adult volunteers infected with *Cryptosporidium* [42].

Calves infected with *C. parvum* had higher serum PCT and NPT levels. Blood PCT levels rapidly increased in response to infection and after the release of CYTs [12,43,44]. Remarkably, higher serum NPT levels reported in this study may trigger a significant cellular immune response in diseased calves. NPT is produced as a result of the cellular immune response when monocytes/macrophages are stimulated by interferon-γ [13]. Increased NPT levels have been described in parasitic conditions [45]. The significant changes in APPs and CYTs levels in *C. parvum*-diseased calves versus those in recovered ones indicated that these makers might be used to estimate treatment success. These markers have been used in both humans and animals for diagnosis, prognosis, and treatment monitoring [7,11,46–48]. There have been several reports of serum OS in sick animals [49,50]. OS can be detected in cells by analyzing changes in antioxidant and oxidant levels [51]. The presence of fat accumulation is strongly influenced by the oxidation process, and the products of lipid peroxidation have been identified as parameters of OS. In body cells, lipid peroxidation

produces MDA, a product of fatty acid peroxidation [52–54]. Interestingly, the serum MDA levels in *C. parvum*-infected calves displayed a specific form of lipid peroxidation and a state of OS. Previous studies have shown that MDA increases in different clinical problems, including sepsis and parasitic and bacterial infection [53,55–58]. As the result of infections or low antioxidant levels, lipid peroxidation is caused by the release of an abundance of free oxygen radicals [59].

Calves infected with *Cryptosporidium* have low concentrations of GSH and SOD due to their considerable role in protecting cells from OS [60]. Consequently, if the GSH and SOD (antioxidants) in the cell cannot remove these radicals, OS may result [61]. Comparable findings were observed in this study as *Cryptosporidium*-infected calves showed higher MDA and diminished GSH and SOD levels when likened to control ones. In calves with *Cryptosporidium* infection, the significant drop in SOD and GSH levels may be due to reduced antioxidant enzyme capacity to neutralize OS. SOD is crucial in catalyzing the radical dismutation process. Previous studies have shown that SOD plays a role in the enhancement of trinitrobenzene sulfonic acid induced colitis [62]. GSH also plays a vital role in preventing cellular damage caused by ROS. Free radicals responsible for the peroxidation of lipids in cell membranes are also responsible for inflicting cellular damage on infected gut cells [63]. Comparable to our results, Abd El-Aziz et al. [64] reported significantly low levels of GSH in rats infected with *C. parvum*. Additionally, Wang et al. [65] reported significantly low levels of GSH and SOD in selenium-deficient C57BL/6 mice, with increased vulnerability of these mice to infection with *C. parvum*. Hence, our results showed that the evaluation of OS status in calves with *Cryptosporidium* infection would evaluate damage caused by free radicals. This study showed a correlation between MDA and each of APPs, PCT, NPT, and CYTs biomarkers. Additionally, we did not ascertain any correlation between MDA and tested antioxidants in calves infected with *C. parvum*. ROC analysis was carried out to differentiate between *C. parvum*-infected and healthy control calves to assess APPs, CYTs, and OS levels. The results of this study showed that all parameters had a high degree of differentiation between *C. parvum*-infected calves and healthy ones [66].

The Youden index was used to determine the optimal threshold for the highest sensitivity, specificity, and accuracy. The supercilious Se, Sp, and accuracy of the APPs, CYTs, and OS markers might be considered a supplemental tool for diagnosing *C. parvum*-infected calves. Because its half-life in the body is caused by renal excretion, NPT remains the biochemically dormant marker [67]. As a result, NPT detection has several advantages over APPs and CYTs detection, as they have a short half-life and rapid decline in serum of infected animals. Our findings demonstrated that PCT, NP, HP, IF-γ, IL-1β, TNF-α, SAA, and OS indicators had a high degree of predictive accuracy for treatment response of *C. parvum*-infected calves at the specified thresholds. Previously, cytokines and APPs were utilized as putative prognostic indicators for various illnesses, indicating therapy efficacy. Comparable findings were reported in previous studies [68–70]. The need for a more extensive clinical assessment of *C. parvum*-infected calves is determined in this study by differences in APPs and CYTs levels in the blood. Likewise, APPs can often be measured as an excellent tool for observing the treatment success in calves with *Cryptosporidium* infection. In this study, PCT and NP were used as a new putative diagnostic and prognostic biomarkers in *C. parvum*-infected calves, and are useful in assessing disease consequences. Our findings are consistent with those previously published in determining the prognostic value of infection using PCT levels [71–73]. In summary, APPs are increasingly being used as biomarkers of inflammation in cattle. There has been evidence that APPs are increased in a variety of metabolic and infectious diseases. Therefore, it is reasonable to conclude that APPs may be a sign of a general inflammatory state of the host rather than of a specific disease (as it is not pathogen-specific). However, the higher levels of APPs perceived in infected calves with *C. parvum* advocates a strong pathogenic role for this protozoan parasite. Consequently, the presence of APPs in the blood of the infected calves suggests a

breach of the immune system by *C. parvum*. Further research is warranted to uncover how APPs contribute to the host's innate immunity.

### 5. Conclusions

*Cryptosporidium* infection was linked to significant alterations in serum PCT, NPT, APPs, CYTs, and OS indicators according to this study. Similarly, this study found that *C. parvum*-infected calves had higher levels of these biomarkers than healthy calves. The findings suggested that the tested biomarkers have a significant role in disease immune pathogenesis. Furthermore, in addition to the clinical examination, measuring APPs, PCT, NPT, CYTs, and OS indicators could be a useful diagnostic and predictive tool for *C. parvum* infection in calves.

**Author Contributions:** Conceptualization, W.E.-D., O.I. and M.F.; methodology, W.E.-D., O.I., M.F., N.A. and A.I.; software, I.E.; validation, W.E.-D., O.I., N.A. and A.A.; formal analysis, I.E.; investigation, W.E.-D.; A.A. and N.A.; resources, W.E.-D., M.F. and O.I.; data curation, W.E.-D., O.I. and M.F.; writing—original draft preparation, W.E.-D.; writing—review and editing, W.E.-D., H.V.M., A.I. and M.F.; visualization, W.E.-D., A.A. and N.A.; supervision, W.E.-D. and O.I.; project administration, W.E.-D.; funding acquisition, W.E.-D. All authors have read and agreed to the published version of the manuscript.

**Funding:** This research was funded by the Deanship of Scientific Research, King Faisal University, Saudi Arabia [Project No. AN00033].

**Institutional Review Board Statement:** Animal Ethics Committee of the College of Veterinary Medicine, King Faisal University, Saudi Arabia, approved all research procedures in this study (Approval number: CVM-2018-0022).

**Informed Consent Statement:** Not applicable.

**Data Availability Statement:** The data presented in this study are available on request from the corresponding author.

**Acknowledgments:** The authors thank the Annual Funding track by the Deanship of Scientific Research, King Faisal University, Saudi Arabia [Project No. AN00033].

**Conflicts of Interest:** The authors declare no conflict of interest.

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
