# Peer review of "Assessment of the Immune Response of Clinically Infected Calves to Cryptosporidium parvum Infection"

_agriculture, doi:10.3390/agriculture12081151_

Round 1

Reviewer 1 Report

1. Thorough proof reading required. Many typos are identified throughout the manuscript.

2. Add ethic guidelines in accordance with the University/institution IACUC approvals involved in the study. 

3. Lines 101-106. Rewrite the sentences to avoid redundancy in information

4. Figure 3 and 4: Explain the X axis. What does "Values" indicate?

5. How Specificity and Sensitivity for each tests calculated/ Explain in the Methods.

6. Describe in the discussion how ROCs contribute to prognostic accuracy?

7. Include in discussion,  the information about the specificity of the biomarkers for Crypto in comparison to other underlying enteric diseases.

Author Response

Dear Dr.

We are pleased to resubmit a revised version of manuscript #1777821# entitled "Assessment of the Immune Response of Clinically Infected Calves to Cryptosporidium Parvum Infection" for Agriculture. Thank you for giving us the opportunity to revise and resubmit this manuscript. We appreciate the time you spent on this manuscript and the detailed and constructive comments that you have provided which have helped us to substantially improve the manuscript. The manuscript has been revised to reflect your comments and suggestions. We look forward to working with you  to move this manuscript closer to publication in Agriculture.

We have responded specifically to each comment/suggestion below and line number was based on the attached version of the manuscript. 

Reviewer 2 Report

Line 84: change “A 202 out of 1,688 calves” for “202 out of 1,688 calves”

Line 95: change “C. parvum infected Calves were treated” for “C. parvum infected calves were treated”

Line 104-105: indicate in italics genus and species

Line 105-107: omit, it is redundant “Fecal samples were diluted 1:1 in dilution buffer and the antigens of Rotavirus, Coronavirus, E. coli, and Cryptosporidium were detected.

Line 131: They do not describe the colorimetric method used and lack reference to it

2.2. Fecal samples examination and 2.3. Molecular confirmation: Do not mention whether they performed the search for Cryptosporidium parvum after treatment (at the time of taking the blood sample)

In figure 3 and 4, indicate for each box the corresponding significance

The references have double numbering

Author Response

(The authors gave the same response as above.)
